# New Regimen of Combining Hepatic Arterial Infusion Chemotherapy and Lipiodol Embolization in Treating Hepatocellular Carcinoma with Main Portal Vein Invasion

**DOI:** 10.3390/jpm13010088

**Published:** 2022-12-29

**Authors:** Ming-Feng Li, Huei-Lung Liang, Chia-Ling Chiang, Wei-Lun Tsai, Wen-Chi Chen, Cheng-Chung Tsai, I-Shu Chen

**Affiliations:** 1Department of Radiology, Kaohsiung Veterans General Hospital, Kaohsiung 813414, Taiwan; 2Department of Medical Imaging and Radiology, Shu-Zen Junior College of Medicine and Management, Kaohsiung 821004, Taiwan; 3Division of Gastroenterology and Hepatology, Department of Medicine, Kaohsiung Veterans General Hospital, Kaohsiung 813414, Taiwan; 4Department of Surgery, Kaohsiung Veterans General Hospital, Kaohsiung 813414, Taiwan

**Keywords:** hepatocellular carcinoma, hepatic arterial infusion chemotherapy, lipiodol embolization, main portal vein invasion

## Abstract

Background: The prognosis of HCC patients with main portal vein invasion (Vp4) is poor. We retrospectively reviewed the therapeutic outcomes with our new HAIC regimen in treating Vp4 HCC patients. Patients and Methods: Seventy-one patients received the new regimen of combining HAIC (daily infusion of cisplatin (10 mg/m^2^), mitomycin-C (2 mg/m^2^) and Leucovorin (15 mg/m^2^) plus 100 mg/m^2^ of 5-fluorouracil (5-FU) using an infusion pump for 5 consecutive days) with Lipiodol embolization between 2002 and 2018. Twenty-two patients (31.0%) also received sorafenib. The Kaplan–Meier curve was used to calculate progression-free survival (PFS) and overall survival (OS). The OS of patients with or without additional sorafenib use or extrahepatic spread (EHS) was also compared. Results: Fifty-six patients (78.9%) had Child-Pugh A liver function. The mean maximal tumor size was 10.3 cm. Twenty patients (28.2%) had EHS at their initial diagnosis. The objective response rate according to the Modified Response Evaluation Criteria in Solid Tumors (mRECIST) and median OS were 64.8% and 13 months. The 1-, 2- and 3-year survival rates were 53.1%, 21.5% and 18.7%, respectively. In the subgroup analysis, there were no significant survival difference between patients with HAIC only vs. HAIC plus sorafenib (14 vs. 13 months) and between patients with vs. without EHS (12 vs. 13 months). Conclusions: Our new HAIC regimen is effective in treating Vp4 HCC patients. Additional sorafenib use with our new HAIC regimen provided no survival benefit.

## 1. Introduction

Portal vein tumor thrombosis (PVTT) is found in approximately 10–40% of all hepatocellular carcinoma (HCC) patients at the time of diagnosis [1,2], and the overall survival (OS) is only 2.7 to 4 months with the best supportive care [3,4,5]. Two randomized placebo-controlled trials for sorafenib (the SHARP and Asia-Pacific trials) have shown that it significantly improves the median OS (10.7 vs. 7.9 months and 6.5 vs. 4.2 months, respectively) in patients with advanced HCC [3,6], with an actual survival benefit of less than 3 months in both Western and Asian populations and an effectiveness that is lower in populations with HCC related to the hepatitis B virus (HBV) [7]. Atezolizumab and bevacizumab combination therapy has recently been recommended as the first-line therapy for advanced HCC [8]. Subgroup analysis showed that this combination therapy was superior to sorafenib in HCC patients with main portal vein tumor thrombus. The median survival time was reported to be 7.6 and 5.5 months, respectively (*p* = 0.104) [9].

In a recent meta-analysis, hepatic artery infusion chemotherapy (HAIC) was found to be more effective in terms of objective response rates (ORR), progression free survival (PFS) and OS than sorafenib for advanced HCC patients [10]. In the literature, the reported median PFS and OS of HCC patients with PVTT treated by HAIC ranged from 2–6 months and 7–10 months, respectively [11,12,13,14,15,16,17,18], with Song et al. reporting the poorest OS (7 months), as 60% of their patients (30/50) presented main portal vein invasion [16]. The results of other studies recommend that HAIC be combined with other therapies, such as radiation therapy [19] or sorafenib [20]. However, objective comparisons of the effectiveness of different treatment regimens have heretofore been precluded by a number of factors, such as the fact that the enrolled patients in most studies were in marked heterogeneity, presenting either PVTT (including segmental (Vp2), lobar (Vp3) or main portal vein (Vp4) invasion) or major PVTT (Vp3 plus Vp4), and that results (such as a 1.54 hazard ratio of Vp3–4 to Vp0–2) were reported in the OS of advanced HCC patients after HAIC [20]. Therefore, it is of significant clinical interest to report solely on Vp4 patients in a large patient series. Moreover, most studies excluded the patients with extrahepatic spread (EHS) for locoregional HAIC therapy; Lyu et al. reported [21] no significant survival difference in HCC patients with or without EHS after HAIC, which accounted for 43% of patients at the initial diagnosis in their series.

In the present retrospective study, we investigated the clinical outcomes of 71 Vp4 HCC patients who were treated by our “new” HAIC regimen, which combined a low dose of HAIC (24-hour infusion of cisplatin, mitomycin C, leucovorin and 5-FU for 5 consecutive days) followed by lipiodol injection, under the hypothesis that synergy between the separate chemo-cytotoxic and embolization effects could be achieved without significantly increasing liver toxicity. Patient subgroups with or without EHS or additional sorafenib target therapy were also analyzed and reported.

## 2. Materials and Methods

The data of consecutive eligible advanced HCC patients for the new regimen of HAIC in our hospital from January 2002 to December 2018 were retrospectively reviewed. HCC was diagnosed by either pathology or the elevation of α-fetal protein (AFP) levels above 400 ng/mL along with at least two different imaging modalities. The inclusion criteria were: (a) aged 18 to 85 years old; (b) advanced HCC with main portal vein invasion (Vp4); (c) Child–Pugh liver function class of A or B; (d) platelet counts ≥50,000/cumm; (e) prothrombin time INR ≤ 1.5; (f) white cell counts ≥2500/cumm; and (g) having received at least 2 courses of HAIC. Patients were excluded based on Eastern Cooperative Oncology Group (ECOG) performance statuses of 3 or 4, or patients who had received concurrent radiotherapy or if radiological follow-up images were unavailable. Patients with prior treatments or with EHS (regional LNs or distant metastasis) were not excluded in this study. The reason for not excluding patients with EHS was that the short-term cause of death in these patients was usually due to intrahepatic tumor rapid progress instead of extrahepatic lesions per se.

### 2.1. Ethics Statement

The study was conducted in accordance with the Declaration of Helsinki and was approved by the ethics review committee of Kaohsiung Veterans General Hospital.

The study was approved by the Kaohsiung Veterans General Hospital Institutional Review Board (IRB No. VGHKS22-CT7-03), which waived informed consent because this was a retrospective study without intervention or obtaining clinical specimens, and all of the data were analyzed anonymously.

### 2.2. New Hepatic Arterial Infusion Chemotherapy (HAIC)

The left subclavian (axillary) artery was punctured under ultrasonographic guidance with the insertion of a 65 cm, 4-F catheter (RC1, Cordis, Johnson-Johnson, Roden, The Netherlands). The gastroduodenal artery (GDA) and/or right gastric artery were occluded by metallic coils to prevent gastroduodenal injury from the anticancer agents. Usually, the 4F catheter was used to embolize the GDA, and a microcatheter was used to embolize the RGA in some patients who revealed prominent RGA. For chemotherapy, the tip of the catheter was placed in the proper hepatic artery or replaced right hepatic artery under fluoroscopic guidance [22]. No permanent delivery port system was used in this study. For HCC with dual hepatic artery blood supply, we did not embolize one of the dual supplying arteries for the purpose of flow diversion. Instead, we performed our HAIC regimen in each of the dual hepatic arteries separately with divided dose. For those with extrahepatic small collateral supplies, we performed superselective tumor embolization by injection of lipiodol and gelfoam pieces as the technique of routine TAE for HCC via the same arterial route.

Our hospital adopted a low-dose HAIC regimen from 1997 for the patients with advanced HCC, with a daily infusion of cisplatin (10 mg/m^2^), mitomycin-C (2 mg/m^2^) and Leucovorin (15 mg/m^2^) (dissolved in 50 mL isotonic sodium chloride solution) for 20–30 min, plus 100 mg/m^2^ of 5-fluorouracil (5-FU, dissolved in 250 mL of isotonic sodium chloride solution) administered for the remainder of the 24 h period using an infusion pump, for 5 consecutive days. Since January of 2002, we added partial embolization therapy by manual slow injection of 10 mL ethiodized oil (Lipiodol, Guerbet, France) into the hepatic artery after the 5 days of chemo-infusion (the “new” HAIC regimen). No gelfoam was injected to occlude the hepatic artery in this combination therapy.

Continuous heparin infusion of 5000 units daily (208 IU/hour) was given to prevent thrombus formation along the catheter surface. The angio-catheter was removed immediately after the injection of Lipiodol. The interval between the 2 courses of treatment was 6 weeks. Three-phase computed tomography (CT) or dynamic magnetic resonance imaging (MRI) of the liver was performed after every 2 courses of treatment, with an upper limit of 6–8 courses or until either radiological progress or death.

### 2.3. Sorafenib Therapy

Patients in the combined HAIC and sorafenib group began sorafenib 400 mg BID either during or after the first HAIC session. Sorafenib doses were reduced, delayed, or temporarily interrupted if it was determined that there was clinically significant toxicity (grade 2 or higher, according to the National Cancer Institute Common Terminology Criteria for Adverse Events, version 4.0) [23]. Dose escalation or rechallenging with sorafenib was resumed when it was determined that toxicity had decreased and that the patient could tolerate the medication well.

### 2.4. Assessment of Response

Radiologic response was determined according to the modified Response Evaluation Criteria in Solid Tumors (mRECIST) guidelines [24] and was defined as follows: complete response (CR)—no evidence of neoplastic disease; partial response (PR)—at least a 30% decrease in the sum of the diameters of “viable” (i.e., enhancement in the arterial phase) target lesions, taking as reference the baseline sum of the diameters of target lesions; stable disease (SD)—any cases that do not qualify for either partial response or progressive disease; and progressive disease (PD)—an increase of at least 20% in the sum of the diameters of viable target lesions, taking as reference the smallest sum of the diameters of viable target lesions recorded since treatment started.

The PFS and OS periods were defined as the time elapsed between treatment initiation and tumor progression (as per the mRECIST guidelines) or death from any cause [24], measured in increments of 0.5 months. As patients with metastatic lesions were not excluded in this study, we assessed the intra-hepatic PFS instead of the whole disease PFS.

### 2.5. Statistical Analysis

The data were expressed as mean ± standard deviation. Categorical variables were compared with the chi-squared test or Fisher’s exact test when appropriate, and continuous variables were compared with the Mann–Whitney test. The median PFS, OS and associated 95% confidence intervals (CI) were analyzed using the Kaplan–Meier curve, and the difference was determined by the log-rank test. Univariate and multivariate analyses of age, sex, tumor size, Child–Pugh classification, AFP level, treatment courses and additional sorafenib treatment were performed using Cox’s regression model with proportional hazards. The therapeutic outcomes in patients with or without additional sorafenib therapy were also compared and analyzed. A *p*-value of less than 0.05 was considered statistically significant. Statistical analysis was performed using the Statistical Product and Service Solutions (SPSS) version 18.0 software (SPSS Inc., Chicago, IL, USA).

## 3. Results

### 3.1. Patients

The enrolled patient flow chart is shown in Figure 1. A total of 60 male and 11 female patients were enrolled in this retrospective study, with a mean age of 60.5 ± 12.3 years (range: 34–81 years). In total, 46 patients (65%) had had hepatitis B infection and 18 patients (25%) had had hepatitis C, while the remaining 7 patients had never contracted either. Sixty patients (84.5%) had Child–Pugh A liver function, while the other eleven patients (15.5%) had Child–Pugh B. Forty-five patients (63%) had AFP ≥ 400 ng/mL. The mean maximal tumor size was 10.3 ± 3.6 cm (range: 3.5–18 cm), with 40 patients (56.3%) having tumor diameters ≥ 10 cm. Forty-two patients (59.2%) had ≥5 tumors, while the other twenty-nine patients had <5. In total, 44 patients (62%) had HCC involvement of the bilateral lobe, and 27 patients had unilateral lobe involvement (23 in the right lobe and 4 in the left lobe). Twenty patients (28.2%) had extra-hepatic spread (EHS) at their initial diagnosis, including lymph node metastasis in thirteen patients, lung metastasis in seven, adrenal gland and pleura metastasis in one each; of these, two patients had both lymph node and lung metastasis. Ten patients (14.1%) had combined hepatic vein/inferior vena cava involvement. In this study, we excluded Child–Pugh C liver function and ECOG >2 patients, thus all the patients were categorized as BCLC stage C. All but 4 patients were beyond the new Milan up-to-7 criteria [25] (i.e., the sum of the size of the largest tumor (in cm) and the number of tumors being greater than 7). A total of 22 patients (31.0%) received additional oral sorafenib therapy during or after the treatment period. The basic demographic data of all 71 patients and the subgroups (with or without sorafenib therapy) are listed in Table 1. There were no statistical differences in age, sex, tumor size, etiology of viral infection, Child–Pugh classification, serum AFP, maximal tumor size, number of tumors, liver involvement, extrahepatic metastasis or combined hepatic vein invasion between the subgroup patients with or without additional sorafenib treatment (Table 1).

### 3.2. Radiologic Response after Treatment

The mean follow-up time was 22.9 ± 29.2 months (range: 3–156 months). The overall response rate was 64.8% (46 patients), with CR in 9.9% (7 patients) and PR in 54.9% (39 patients). As for dividing the patient groups into those with or without sorafenib therapy, the ORR were 67.3% (CR + PR = 12.2% + 55.1%) in the without sorafenib group and 59.1% (CR (Figure 2) + PR = 4.5% + 54.6%) in the with sorafenib group, respectively (*p* = 0.501). The radiologic response rates of these two patient groups are presented in Table 2.

### 3.3. Progression-Free and Overall Survival Analyses

The median PFS of the 71 patients was 9 months (95% confidence interval: 5.4–12.6 months). The median OS was 13 months (95% confidence interval: 10.5–15.5 months), with 17 months for the responders and 6 months for the non-responders. The median OSs of patients with and without EHS were 13 and 12 months, respectively (*p* = 0.434). As to the liver function status, the median OS was 14 months for Child–Pugh A patients and 6 months for Child–Pugh B patients. The overall 1-, 2- and 3-year survival rates were 53.1%, 21.5% and 18.7%, respectively (Figure 3).

### 3.4. Univariate and Multivariate Analyses

Univariate survival analysis using the Kaplan–Meier method and the log-rank test showed a statistically significant survival benefit in patients with a liver function of Child–Pugh A (median OS: 14 months vs. 6 months for Child–Pugh B patients, *p* < 0.01) and a tumor size of 10 cm or less (median OS: 15 months vs. 9 months for >10 cm, *p* = 0.046) and who showed a therapeutic response (median OS: 17 months for responders vs. 6 months for non-responders, *p* < 0.01) (Table 3). Multivariate Cox regression analysis revealed that the significant factors with adjusted hazard ratios (HR) for OS were Child–Pugh classification (HR: 2.30, 95% CI: 1.18–4.49, *p* = 0.015), maximal tumor size (HR: 1.77, 95% CI: 1.00–3.12, *p* = 0.049) and therapeutic response (HR: 8.66, 95% CI: 3.94–19.07, *p* < 0.01) (Table 3).

### 3.5. Subgroup Analyses

The objective response rate (ORR) was 67.3% (CR: 12.2% + PR: 55.1%) in the subgroup patients (*n* = 49) with the new HAIC regimen only and 59.1% (CR: 4.5% and PR: 54.5%) in patients (*n* = 22) who had received sorafenib plus therapy. The PFS and OS of the subgroup patients were 9 months (95% confidence interval: 3–14.5 months) and 14 months (95% confidence interval: 9.2–18.8 months) in the former, and 9 months (95% confidence interval: 3.4–14.6 months) and 13 months (95% confidence interval: 9.6–16.4 months) in the latter (Figure 4). There were no significant differences in ORR, PFS and OS between these two groups of patients. Using Cox’s proportional hazard model, we investigated the factors related to survival. Except for better survival in patients without hepatic vein/IVC invasion in the without sorafenib group (*p* = 0.027), no other factors were related to OS in the univariate and multivariate analyses of our study population (Figure 5). Additional propensity score matching analyses of the baseline characteristics was tried between these two groups but failed to balance well.

### 3.6. Major Complications

None of the patients died due to immediate HAIC complications. Any adverse events suffered by the patients in this study are described in accordance with the National Cancer Institute Common Terminology Criteria for Adverse Events (NCI-CTC AE) grading system. Sixteen patients developed fever during HAIC, and among them, four developed bacteremia and were treated successfully by antibiotics. One patient developed an overt subcutaneous hematoma at the puncture site but recovered soon and did not require further management. No vascular complications were found, including occlusion or vasculitis of the hepatic artery. No overt adverse effects were observed in the sorafenib plus therapy patients.

## 4. Discussion

Portal vein tumor thrombosis (PVTT) occurs in 35–50% of HCC patients and involves the main trunk at the time of diagnosis in 15–30% of cases [1,2]. Although the HCC treatment guidelines that are endorsed by the American Association for the Study of Liver Diseases [26] (AASLD) and European Association for the Study of the Liver [27] (EASL) recommend sorafenib therapy as the sole treatment for advanced HCC, the clinical benefit of sorafenib by itself has been shown to be marginal [6,28]. Two randomized clinical trials reported the use of sorafenib prolonged the median overall survival relative to placebo control in patients with macroscopic vascular invasion from 4.9 months to 8.1 months (subanalysis of the SHARP trial) [28] and from 4.2 months to 6.5 months (the Asia-Pacific trial) [6]. However, the efficacy of sorafenib has been shown to be lower in areas where HBV shows predominance, such as in Asia (70%) [7], a finding that was confirmed by the GIDEON (Global Investigation of Therapeutic Decisions in Hepatocellular Carcinoma and of Its Treatment with Sorafenib) study, suggesting that worsened prognosis factors have become more common in Asian populations [29]. In recent years, there have been remarkable advances in systemic treatments for HCC. Cheng et al. reported a promising clinical outcome for a combination therapy of atezolizumab and bevacizumab, with a better overall survival (19.2 months) than that of sorafenib alone (13.4 months) in Child A, unresectable HCC patients [30]. It is worth noting that 61.6% and 57.0% patients in each of their groups had no macrovascular invasion at study entry. Since vascular invasion in the advanced HCC patients is a prognostic risk of OS, and the OS of sorafenib in a prior Asian randomized controlled trial study [6] was only 6.5 months (compared with the OS of 13.4 months of the afore-mentioned study [30]), we speculated that the patient status of enrollment may be not the same in the two studies, as the definition of unresectable HCC may cover a wide variety of the disease status. This phenomenon again emphasizes the importance of clearly stratifying HCC status, as we did with the Vp4 patients in our present study. In addition, this high medical cost of combination regimen may limit its extensive application in advanced HCC. Searching for alternative-yet-effective therapies for the macrovascular-invaded HCC patients is therefore warranted.

With refinements in surgical techniques, patient selection and perioperative management, resection for HCC with tumor thrombosis extending to the major portal vein is now possible. Chok et al. reported median OS of 8.58 months in patients with PVTT extending to or beyond the portal vein bifurcation treated by partial hepatectomy with thrombectomy [31]. In a review article, Nevarez et al. concluded that partial hepatectomy with *en bloc* resection of PVTT in second-order and distal portal branches could offer significant benefits in terms of outcome measures in carefully selected patients by experienced centers, particularly in the sequelae of portal hypertension [32]. Lee et al. also reported that PVTT extending to the main portal vein was a significant risk factor for recurrence and worse OS (*p* < 0.01 for both) in patients who underwent living donor liver transplants [33]. Given the development of effective systemic and locoregional therapy options, surgical resection in HCC patients with advanced PVTT should be cautioned against [32].

Im et al. conducted a nation-wide, multicenter study of radiotherapy (RT) for HCC patients with major PVTT and reported that the median OS of RT alone versus combined with other treatments were 8.7 months and 10.4 months, respectively (*p* = 0.023) [19]. Katamura et al. reported a matched case-control study of a combination therapy of HAIC with three-dimensional conformal RT to treat HCC patients with major PVTT. Although the ORR was significantly higher for the RT group (75% vs. 25%, *p* = 0.012), there were no significant differences between the median OS (7.5 vs. 7.9 months) [34]. Yttrium-90 radioembolization particles, especially of the glass microspheres, has a very small particle size (30 mm), allowing for deep infiltration into the tumor without ischemia of the hepatic parenchyma [35]. The absence of overt arterial embolization effect is important for patients who have already had their portal venous occlusion due to malignant tumor invasion, as complete loss of blood supply from both the hepatic arterial and portal venous flow may result in very unfavorable clinical outcomes. Yttrium—90 Transarterial Radioembolization (TARE) was reported to be superior to sorafenib in prolonging survival of PVTT HCC patients in some retrospective studies [33,34,35,36,37,38,39]. Abouchaleh et al. reported that 77 Vp4 patients receiving TARE had a median OS of 7.7 months in Child–Pugh A or B7 patients and 3.4 months for ≥B8 [40]. Two Phase III trials—Sorafenib versus Radioembolization in Advanced Hepatocellular carcinoma (SARAH) [41] and Selective Internal Radiation Therapy Versus Sorafeni (SIRveNIB) [42]—failed to demonstrate any significant superiority of TARE compared with sorafenib. In the SARAH study, the presence of PVTT seemed to favor sorafenib in term of OS.

In Japan and South Korea, HAIC is recommended instead of sorafenib as the first-line treatment option for advanced patients with PVTT [43]. A recent meta-analysis showed that HAIC was more beneficial than sorafenib for advanced HCC patients in terms of ORR, PFS and OS [9]. The reported tumor ORR and median OS ranged from 28% to 48% and 5.7 to 10.5 months, based on the enrolled patient characteristics, with either cisplatin-based [15,16,17] or oxaliplatin-based regimens [20]. Jeong [13], Song [44] and Ahn [45] compared the therapeutic efficacy of HAIC alone and with sorafenib in treating advanced HCC (including Child–Pugh A and B patients) and found that patient survival could be significantly increased from 4.9–6.4 months to 7.1–10.0 months, respectively. Nagai et al., in a small series study (34 patients), reported that HAIC plus sorafenib treatment had more clinical benefit for PVTT patients than HAIC alone in Child–Pugh A patients (315 vs. 197 days) but not in Child–Pugh class B patients (234 vs. 228 days) [46]. Liang further confirmed this conclusion in a large series of 225 patients, with a better OS of 12.9 months in the HAIC plus sorafenib group versus 10.5 months in the HAIC alone group, with a hazard ratio of 1.54 of Vp3–4 vs. Vp0–2 (*p* = 0.007) [20]. In the present study, the median OSs of our 71 Vp4 HCC patients and 56 Child–Pugh A liver function patients were 13 and 14 months, respectively, which were superior to those of the RT, TARE, HAIC alone and HAIC plus RT or sorafenib combination therapy patients in other studies. Table 4 compares the therapeutic outcomes of previously reported multidisciplinary treatment of advanced liver cancer with portal vein tumor thrombus.

Iwamoto et al. reported a multicenter study of a new FP HAIC regimen (cisplatin suspended in lipiodol combined with 5-fluorouracil) to treat advanced HCC [47]. After propensity score matching, the new-FP HAIC regimen revealed a better median OS than that of sorafenib (12 vs. 7.9 months, *p* < 0.001). They attributed the better results to two possible reasons; first, lipiodol induces vascular embolization, and second, the use of anticancer drugs in lipiodol suspension prolongs their retention in the target tissues, allowing their continuous release in tumor tissues [48]. In proposing our new HAIC regimen, we hypothesized that although varying degrees of cellular damage will inevitably ensue after continuous chemoinfusion, injured cancer cells can recover during the interval time between treatment courses or cycles. Therefore, an additional embolic effect by injecting lipiodol may help induce cellular death by some synergistic mechanism. Moreover, because of the high affinity of lipiodol for tumors, most of the injected 10 mL lipiodol will aggregate and be retained in the HCC mass, leaving the normal liver parenchyma less damaged. As the major portal veins were already occluded in these advanced HCC patients, we did not inject gelfoam pieces for embolization in order to preserve hepatic artery patency. The former hypothesis was confirmed by the clinical fact that the ORR increased from 20% of our prior sole low-dose HAIC study [22] to 64.8% with a median OS of 13 months in our Vp4 patients.

Although HCC patients with EHS are usually considered not candidates for locoregional therapy, Lyu et al. reported their experience of HAIC in 116 patients with 43% (50 patients) presenting with EHS^21^. Their results reveal a trend of better median OS in patients without EHS than those with EHS (14.8 vs. 9.8 months) but without reaching significance (*p* = 0.068). In Iwamoto’s series [47], HAIC therapy did not have less OS in patients with MVI and EHS as compared with sorafenib therapy (7 and 5 months, respectively, *p* = 0.28). In the present series, 20 of the 71 (28.2%) Vp4 patients presented with EHS. Our results also reveal no significant survival difference between these two group with or without EHS (12 vs. 13 months, *p* = 0.434). The median OS of the patients with EHS after HAIC was superior to those of supportive care (2.7–4 months) [3,4,5] or target/immunotherapy (5.5–7.6 months) [9]. These findings might suggest that EHS should not be an absolute contraindication of HAIC in treating advanced HCC patients.

In the present study, we failed to demonstrate the clinical benefits of additional sorafenib use. One possible explanation is that the synergistic effect by adding the lipiodol embolization far outweighs the stabilization effect of sorafenib. Another possible explanation may be that the with sorafenib patient group had higher percentages of high AFP (≥400 ng/mL) levels, although not reaching statistical significance, which have been shown to be an adverse prognostic factor for mortality in patients with advanced HCC [20], although not in our series of the 71 Vp4 patients. The above factors may have influenced the OS of the sorafenib-plus group in our patients. Additional prospective comparative studies are warranted to draw a definitive conclusion on this issue.

## 5. Limitations

There were some limitations in this study. First, this was a retrospective study, which was affected by baseline confounding factors. Second, as patients without post-treatment imaging follow-up were excluded, this may have skewed the OS in a favorable direction. Third, the treatment adverse effects might not be thoroughly reported via medical chart review in a retrospective study. Finally, in the subgroup analysis, the patients’ demographic data between groups failed to be well-balanced for propensity score matching.

## 6. Conclusions

Our new HAIC regimen, which combines arterial chemoinfusion and lipiodol embolization, is effective in treating HCC patients with main portal vein invasion. It may be adopted as the first-line therapy for these patients even with EHS, especially in HBV-predominant regions of the world, in which sorafenib target therapies commonly show worsened prognoses.

## Figures and Tables

**Figure 1 jpm-13-00088-f001:**
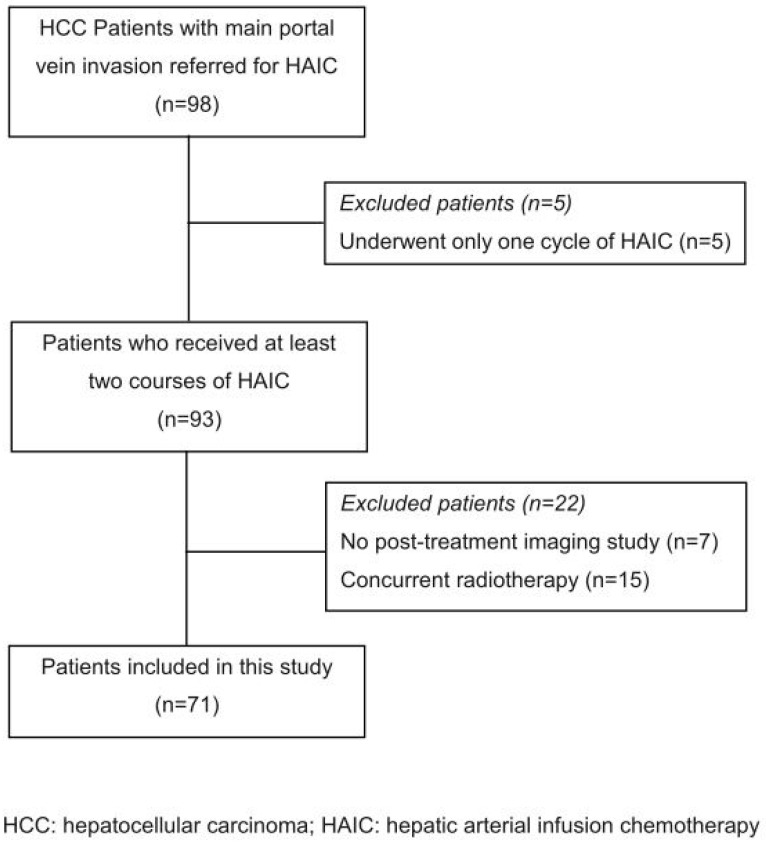
Flow chart of patient inclusion and exclusion.

**Figure 2 jpm-13-00088-f002:**
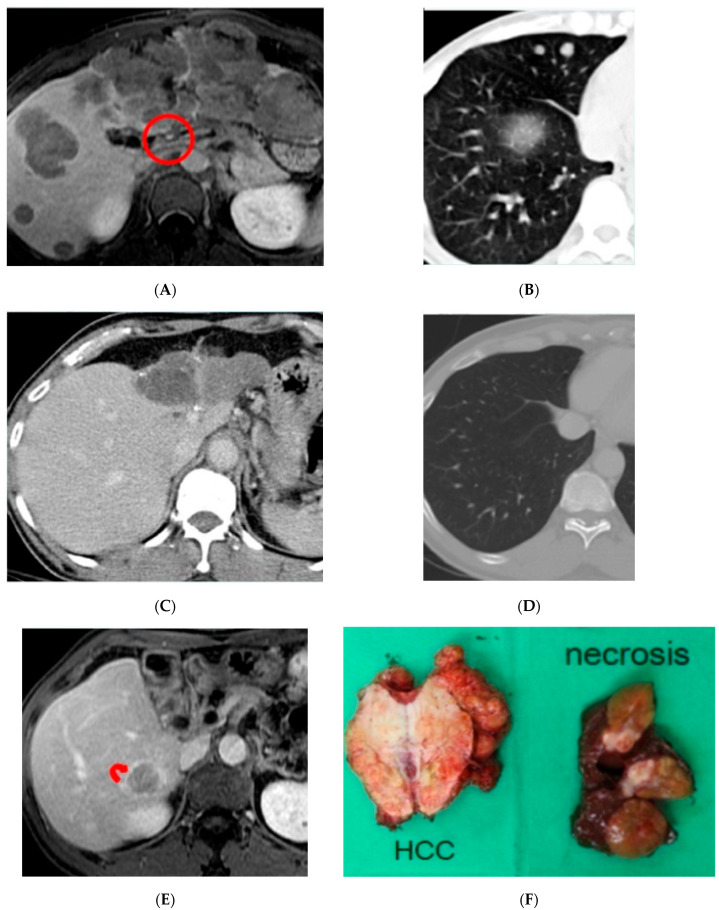
A 50 y/o male with multiple HCC in the bilateral lobe of liver. The largest tumor was 14 cm in the left lobe liver. (**A**). Liver MRI showing tumor invasion into the MPV (circle). (**B**). Chest CT also showing metastatic nodules in the right lung. (**C**). Liver CT showing complete remission after 6 courses of HAIC plus sorafenib therapy. (**D**). Chest CT also demonstrating disappearance of the lung metastatic lesions. (**E**). Liver MRI revealing a new recurrent 3.5 cm HCC (arrow) in the right lobe liver 20 months after the last treatment. (**F**). Surgical specimen showing the recurrent HCC in the right lobe liver and complete necrosis of the atrophic left lobe lesions. At the time of publication, this patient had survived for more than 67 months.

**Figure 3 jpm-13-00088-f003:**
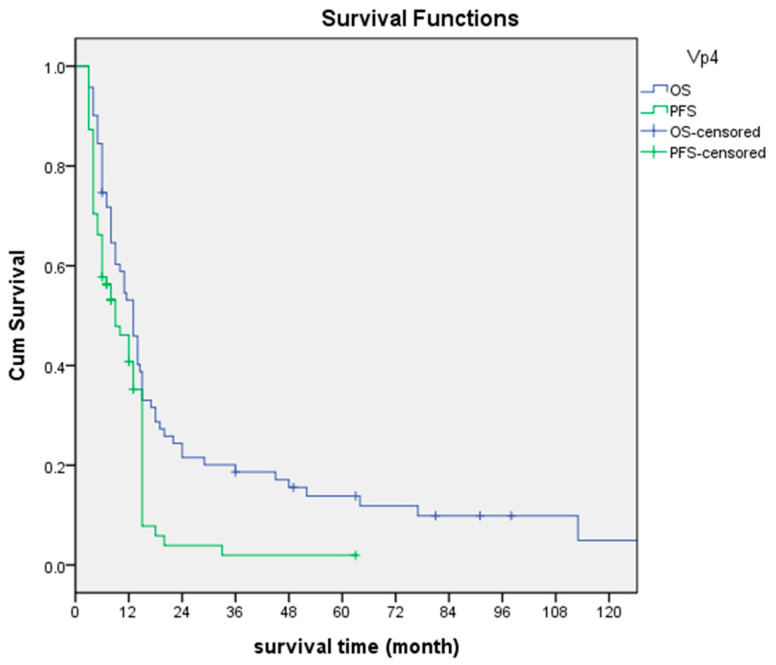
PFS and OS of all 71 Vp4 patients.

**Figure 4 jpm-13-00088-f004:**
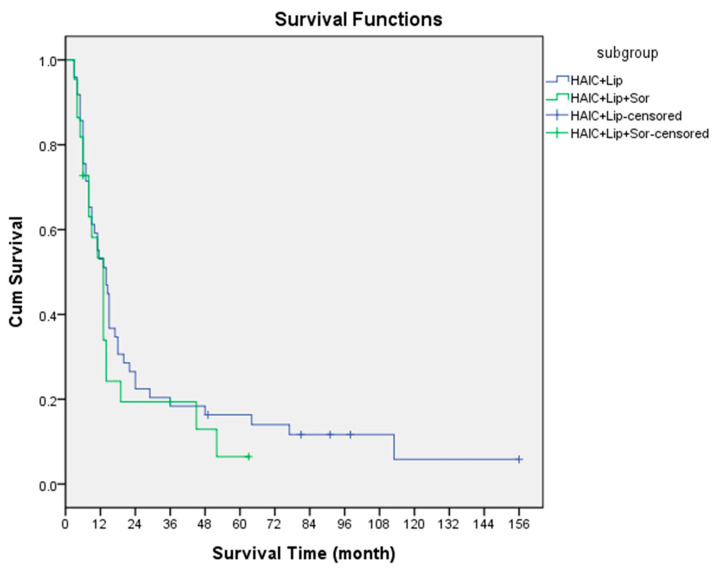
OS of the with and without sorafenib therapy patient subgroups.

**Figure 5 jpm-13-00088-f005:**
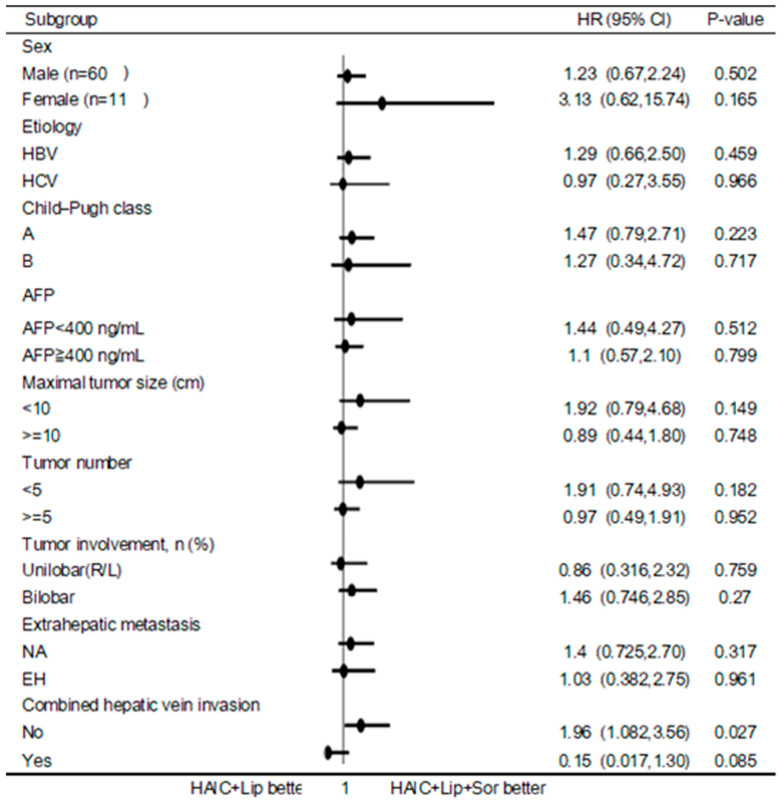
Forest plot of the treatment effect on OS in the subgroup analysis. Horizontal lines represent 95% confidence interval. The position of the circle represents the point estimate of the treatment effect.

**Table 1 jpm-13-00088-t001:** Basic demographic data of the 71 Vp4 HCC patients.

Variable	Overall	HAIC + Lip	HAIC + Lip + Sor	*p* Value
Patients	71	49	22	
Age (years old, mean ± SD)	60.5 ± 12.3	61.9 ± 11.3	57.4 ± 14.1	0.22
Male/Female, *n* (%)	60/11 (84.5/15.5)	41/8 (83.7/16.3)	19/3 (86.4/13.6)	0.60
Etiology, *n* (%)				
HBV/HCV/non-BC	46/18/7 (65/25/10)	31/14/4 (63/29/8)	15/4/3 (68/18/14)	0.96
Child–Pugh score, *n* (%)				
A/B	60/11 (84.5/15.5)	39/10 (79.6/20.4)	21/1 (95.5/0.05)	0.29
AFP ≥ 400 ng/mL, *n* (%)	45 (63.0)	27 (55.1)	18 (81.8)	0.17
Maximal tumor size (cm)	10.3 ± 3.6	9.9 ± 3.7	11.3 ± 3.4	0.57
<10	31 (43.7)	24 (49.0)	7 (31.8)	
≥10	40 (56.3)	25 (51.0)	15 (68.2)	
Tumor number				0.45
≥5	42 (59.2)	27 (55.1)	15 (68.2)	
<5	29 (40.8)	22 (44.9)	7 (31.8)	
Tumor involvement, *n* (%)				0.30
Unilobar (R/L)	23/4 (32.4/5.6)	17/4 (34.7/8.2)	6/0 (27.3/0)	
Bilobar	44 (62.0)	28 (57.1)	16 (72.7)	
Extrahepatic metastasis	20 (28.2)	12 (24.5)	8 (36.4)	0.62
Combined HV invasion	10 (14)	5 (10.2)	5 (22.7)	0.77
Treatment courses	3.96 ± 1.84	3.69 ± 1.61	3.64 ± 1.68	0.891

HAIC + Lip: hepatic arterial infusion chemotherapy plus lipiodol injection; HAIC + Lip + Sor: hepatic arterial infusion chemotherapy plus lipiodol injection plus sorafenib

**Table 2 jpm-13-00088-t002:** Summary of best response based on the mRECIST criteria.

	Overall Response		
	HAIC Group (%)	SoraHAIC Group (%)	*p*
CR	6 (12.2)	1 (4.5)	0.001
PR	27 (55.1)	12 (54.6)	0.933
SD	7 (14.3)	3 (13.6)	0.886
PD	9 (18.4)	6 (27.3)	0.114
DCR	40 (81.6)	16 (72.7)	0.114
ORR	33 (67.3)	13 (59.1)	0.501

**Table 3 jpm-13-00088-t003:** Univariate and multivariate analysis of the hazard risk on OS of the 71 Vp4 patients.

	Univariate Analysis		Multivariate Analysis	
Variable	HR (95% CI)	*p* Value	HR (95% CI)	*p* Value
Sex	1.47 (0.76, 2.87)	0.26		
Etiology (B/C)	0.73 (0.41, 1.30)	0.28		
Child–Pugh class (A/B)	3.03 (1.66, 5.53)	<0.01 *	2.30 (1.18, 4.49)	0.015 *
AFP (<400 ng/mL/≥ 400 ng/mL)	1.45 (0.86, 2.43)	0.16		
Maximal tumor size (cm) (<10/≥10)	1.69 (1.01, 2.82)	0.046 *	1.77 (1.00, 3.12)	0.049 *
Tumor number (<5/≥5)	1.23 (0.74, 2.06)	0.42		
Tumor involvement (Unilobar/Bilobar)	1.07 (0.65, 1.78)	0.79		
Extrahepatic metastasis (No/Yes)	0.95 (0.56, 1.64)	0.87		
Combined hepatic vein invasion (No/Yes)	0.91 (0.41, 2.00)	0.81		
Therapeutic Response (CR + PR/SD + PD)	7.30 (3.73, 14.28)	<0.01 *	8.66 (3.94, 19.07)	<0.01 *

* *p* < 0.05.

**Table 4 jpm-13-00088-t004:** Comparison of clinical outcomes treated by different modalities for HCC patients with major PV invasion.

Author	Year	Pt No.	Treatment	Vp3/4	ORR	PFS	OS (m)
Kaneko [11]	2002	29	HAIC	65/35%	45%	NA	8
Niizeki [15]	2012	71	HAIC	69/31%	35%	NA	10.2
Bruix [27]	2012	108	Sor vs. placebo	MVI	DCR: 38.9 vs. 26.8%	4.1 vs. 2.7	8.1/4.9
Song [16]	2013	50	HAIC	26/60%	32%	2	7
Nagamatsu [45]	2016	52	HAIC	25/29%	75%	8.6	27/12 (Vp4)
Moriguchi [18]	2017	32	HAIC	78/22%	31%	3.7	10.3
Im [19]	2017	986	RT + TACE/HAIC	51/49%	52%	NA	10.2
Abouchaleh [38]	2018	77	Y-90 SIRT	Vp4	NA	NA	5
Ahn [43]	2021	35/38	Sor vs. HAIC	Vp4	0 vs. 5.2%	2.1 vs. 6.2	6.4 vs. 10
Liang [20]	2021	93	HAIC + Sor	65%	37.4%	7	12.9
current study		71	HAIC	Vp4	64.8%	9	13

Sor, sorafenib; HAIC, hepatic arterial infusion chemotherapy; TACE, transcatheter arterial chemoembolization; Y-90 SIRT, Yttrium-90 selective internal radioembolization; Vp3, lobar portal vein invasion; Vp4, main portal vein invasion; MVI, macrovascular invasion; ORR, objective response rate; DCR, disease-controlled rate; NA, not applicable; PFS, progression free survival; OS, overall survival.

## Data Availability

None.

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
