# Peer review of "New Regimen of Combining Hepatic Arterial Infusion Chemotherapy and Lipiodol Embolization in Treating Hepatocellular Carcinoma with Main Portal Vein Invasion"

_jpm, 2022, doi:10.3390/jpm13010088_

Round 1

Reviewer 1 Report

well designed study, interesting topic and with important clinical relevance. I would have emphasized the difference with the tare in the literature more.

clinical experience and scientific studies show that sorafenib should never be administered in close proximity to vascular procedures, the absence of complications is a factor that should be adequately emphasised.

Author Response

Thank you for your valuable comments.

Followings are our replies:  

  1. well designed study, interesting topic and with important clinical relevance. I would have emphasized the difference with the tare in the literature more.
        Reply: We add a short discussion of TARE in the 3rd paragraph of the    

 “Discussion”. “Yttrium-90 radioembolization particles, especially of the glass     

microspheres, has a very small particle size (30 mm), allowing for deep

infiltration into the tumor without ischemia of the hepatic parenchyma (35).

The absence of overt arterial embolization effect is important for patients who have already had their portal venous occlusion due to malignant tumor

invasion as complete loss of blood supply from both the hepatic arterial and
portal venous flow may result in very unfavorable clinical outcomes”.
As to the therapeutic outcome of TARE, we consider that the report of

Abouchaleh’s series is representative as they reported the largest series of

Vp4 patients receiving TARE in the literature.

  1. clinical experience and scientific studies show that sorafenib should never be administered in close proximity to vascular procedures, the absence of complications is a factor that should be adequately emphasised.
    Reply: Yes, some anti-angiogenic agents (e.g. Avastin) will suppress VEGF, which     may decrease the therapeutic effect or even causing complications during vascular interventional procedures. In our hospital, VEGF inhibitors will be discontinued for at least 2-4 weeks for patients who are going to receive TARE. But for tyrosine kinase inhibitors, we don’t hold sorafenib therapy in either HAIC or TARE. We add a statement in the last paragraph of the “Results”- No overt adverse effects were observed in the sorafenib plus therapy patients.

Reviewer 2 Report

Abstract:

1. "oral sorafenib" should be corrected into "sorafenib", because sorafenib is usually oral-administrated. 

1. "71 patients, 22 patients" should be corrected into "Seventy-one patients, Twenty-two patients" because they are located at the front of the sentences. 

2. "78.9% patients, 28.2% patients" --> "56 patients (78.9%), 20 patients (28.2%). 

3. " The Modified Response Evaluation Criteria in Solid Tumors (mRECIST) of objective response rate " --> The objective response rate according to the Modified Response Evaluation Criteria in Solid Tumors (mRECIST) and median OS were ,,,,,,

4. "or patients without vs. with EHS (p=0.434)" --> "and between patients with vs. without EHS (p=0.434)"

5. You'd better decide whether to use period (months) or p value in the subgroup analysis. 

6. "no extra clinical benefit" --> "no survival benefit" 

7. The authors should describe "what is the HAIC new regimen?" briefly in Patients and Methods, including the name of all agents used. 

Introduction:

1.  "The overall survival (OS) is only 2.7 to 4 months, even with the best supportive care." --> "Portal vein tumor thrombosis (PVTT) is found in approximately 10 - 40% of all hepatocellular carcinoma (HCC) patients at the time of diagnosis [1-2] and the overall survival (OS) is only 2.7 to 4 months with the best supportive care [3-5]."

2. This introduction consists of one paragraph. I recommend you divide the paragraph at "(p = 0.104) [9]. V In a recent meta-analysis..", " in their series. V In the present retrospective study,,," Three paragraphs would be better. 

Materials and Methods:

1. This technical description for HAIC catheter insertion is too brief. The authors should additionally describe 1) the kind of infusion port with the brand name, 2) how to manage the extrahepatic collaterals 3) how to manage the dual hepatic artery supply 

2. "Our hospital adopted a low dose HAIC regimen in1997 for cirrhotic patients" --> "Our hospital adopted a low dose HAIC regimen from 1997 for the patients with advanced HCC

3. Does your "new" HAIC regimen mean the combination of the previous chemotherapeutic agents' regimen plus additional Lipiodol embolization? So, the chemotherapeutic agents' regimen itself did not change, right?

4. What is "partial" embolization therapy? Does it mean incomplete embolization of Lipiodol (not enough for complete lipiodol accumulation for each tumor) or just lipiodol embolization (lipiodolization) only without additional particles? 

5. 10 mL of Lipiodol was used in this study. But the same dose of Lipiodol could not be used in every patient in real practice. Please, describe the exact amount of Lipiodol used (x ~xx mL, mean x mL) and clearly mention the endpoint of Lipiodol injection. 

6. " The angio-catheter was removed immediately after the injection of Lipiodol.": This means you punctured additional arterial access for insertion of the angio-catheter into the hepatic artery for Lipiodol embolization, right? The authors should briefly describe this procedure, including the use of microcatheter. 

7. "The interval between the 2 courses of treatment was 6 weeks." Does this mean that the interval between each course of treatment was 6 weeks?  

8. Was the upper limit of courses 6 or 8? 

Results:

1. All numbers at the beginning of a sentence should be written as letters. 

2.  The arrow markers in Figure 2 are not usual. In Figure 2A, the IVC invasion is not definite. Did this patient receive liver transplantation? The authors should mention about operation method. The photos of the GB and its stones are not necessary. Please, crop them out from Figure 2G. 

Discussion:

1. "In Japan, HAIC is recommended instead of sorafenib as the first-line treatment" --> "In Japan and South Korea,," because many recent reports were published in South Korea. 

2.  "Another possible explanation may be that the with-sorafenib patient group had higher percentages of high AFP (≥400ng/mL) levels," was not compatible with P=0.17. This description is not statistical. 

Conclusion:

1. "The study was conducted according to the guidelines of the Declaration of Helsinki, ..." is not necessary for Conclusion.  

Author Response

Thank you for your valuable comments.

Followings are our replies:

Abstract:

  1. "oral sorafenib" should be corrected into "sorafenib", because sorafenib is usually
    oral-administrated. 
    Reply: revised as per your suggestion.
  2. "71 patients, 22 patients" should be corrected into "Seventy-one patients,  

    Twenty-two patients" because they are located at the front of the sentences. 
    Reply: revised as per your suggestion

  1. "78.9% patients, 28.2% patients" --> "56 patients (78.9%), 20 patients (28.2%). 

    Reply: revised as per your suggestion

  1. " The Modified Response Evaluation Criteria in Solid Tumors (mRECIST) of objective response rate " --> The objective response rate according to the Modified Response Evaluation Criteria in Solid Tumors (mRECIST) and median OS were ,,,,,,
    Reply: revised as per your suggestion
  2. "or patients without vs. with EHS (p=0.434)" --> "and between patients with vs. without EHS (p=0.434)"
    Reply: revised as per your suggestion
  3. You'd better decide whether to use period (months) or p value in the subgroup analysis. 
    Reply: We use “months” in both results.
  4. "no extra clinical benefit" --> "no survival benefit" 
    Reply: revised as per your suggestion
  5. The authors should describe "what is the HAIC new regimen?" briefly in Patients and Methods, including the name of all agents used. 
    Reply: We had briefly described the HAIC regimen in “Patients and Methods”.

.

Introduction:

  1. "The overall survival (OS) is only 2.7 to 4 months, even with the best supportive " --> "Portal vein tumor thrombosis (PVTT) is found in approximately 10 - 40% of all hepatocellular carcinoma (HCC) patients at the time of diagnosis [1-2] and the overall survival (OS) is only 2.7 to 4 months with the best supportive care [3-5]."
    Reply: revised as per your suggestion.
  2. This introduction consists of one paragraph. I recommend you divide the paragraph at "(p = 0.104) [9]. V In a recent meta-analysis..", " in their series. V In the present retrospective study,,," Three paragraphs would be better.

   Reply: revised as per your suggestion. 

Materials and Methods:

  1. This technical description for HAIC catheter insertion is too brief. The authors should additionally describe 1) the kind of infusion port with the brand name, 2) how to manage the extrahepatic collaterals 3) how to manage the dual hepatic artery supply 
    Reply: Sorry for not clear description of our HAIC technique.
    1) We use a 4F temporary angio-catheter for chemoinfusion via the left subclavian route without infusion port implantation in our patients-added in the “New Hepatic Arterial Infusion Chemotherapy” paragraph.
    2-3). We detailed the technique in the “New Hepatic Arterial Infusion Chemotherapy” paragraph as “For HCC with dual hepatic artery blood supply, we didn’t embolize one of the dual supplying artery for the purpose of flow diversion. Instead, we perform our HAIC regimen in each of the dual hepatic artery separately with divided dose. For those with small extrahepatic collateral supplies, we performed superselective tumor embolization by injection of lipiodol and gelfoam pieces as the technique of routine TAE for HCC via the same arterial route.”
  2. "Our hospital adopted a low dose HAIC regimen in1997 for cirrhotic patients" --> "Our hospital adopted a low dose HAIC regimen from 1997 for the patients with advanced HCC" 
    Reply: revised as per your suggestion.
  3. Does your "new" HAIC regimen mean the combination of the previous chemotherapeutic agents' regimen plus additional Lipiodol embolization? So, the chemotherapeutic agents' regimen itself did not change, right?

    Reply: Yes, it is.

  1. What is "partial" embolization therapy? Does it mean incomplete embolization of Lipiodol (not enough for complete lipiodol accumulation for each tumor) or just lipiodol embolization (lipiodolization) only without additional particles? 
    Reply: Yes, both are correct. In cTACE, 10 ml lipiodol is inadequate for a large advanced HCC as in our patients of this study. Meanwhile, we don’t inject gelfoam or other embolizers after lipiodol injection. Thereafter, we say it “partial embolization”.
  2. 10 mL of Lipiodol was used in this study. But the same dose of Lipiodol could not be used in every patient in real practice. Please, describe the exact amount of Lipiodol used (x ~xx mL, mean x mL) and clearly mention the endpoint of Lipiodol injection. 

Reply: In our new HAIC regimen, we supposed the injection of lipiodol (partial  

    embolization) played an adjunctive effect of chemoinfusion. Thereafter, we injected  

    a fixed dose of 10ml lipiodol in every case of our patients.

  1. " The angio-catheter was removed immediately after the injection of Lipiodol.": This means you punctured additional arterial access for insertion of the angio-catheter into the hepatic artery for Lipiodol embolization, right? The authors should briefly describe this procedure, including the use of microcatheter. 
    Reply: The infusion of chemo-agents and the injection of lipiodol were both performed via a same 4F angio-catheter placed in the common or proper hepatic artery.
  2. "The interval between the 2 courses of treatment was 6 weeks." Does this mean that the interval between each course of treatment was 6 weeks?  

    Reply: Yes, it is.

  1. Was the upper limit of courses 6 or 8? 
    Reply: Generally speaking, from an oncologist point of view, if a chemo-regimen fails to achieve complete remission after 6 courses of treatment, it means some of the tumor cells are chemo-resistant. More courses of the same regimen treatment usually result in no extra-benefit clinically. Thereafter, we set the upper limit of treatment courses of 6. But as this was a retrospective review, we did find some patients having undergone 8 courses of treatments.

Results:

  1. All numbers at the beginning of a sentence should be written as letters. 

Reply: revised as per your suggestion.

  1. The arrow markers in Figure 2 are not usual. In Figure 2A, the IVC invasion is not definite. Did this patient receive liver transplantation? The authors should mention about operation method. The photos of the GB and its stones are not necessary. Please, crop them out from Figure 2G. 
    Reply: We delete Fig.2A. This patient underwent left hepatectomy and partial segmentectomy of the right lobe without transplantation. A new photo of Figure 2G without surgical specimen of GB was presented.

Discussion:

  1. "In Japan, HAIC is recommended instead of sorafenib as the first-line treatment" --> "In Japan and South Korea,," because many recent reports were published in South Korea. 
    Reply: revised as per your suggestion. 
  2. "Another possible explanation may be that the with-sorafenib patient group had higher percentages of high AFP (≥400ng/mL) levels," was not compatible with P=0.17. This description is not statistical. 

   Reply: We add “although not reaching statistical significance” after “Another

   possible explanation may be that the with-sorafenib patient group had higher   

   percentages of high AFP (≥400ng/mL) levels.

Conclusion:

  1. "The study was Declaration according to the guidelines of the Declaration of Helsinki, ..." is not necessary for Conclusion.  
    Reply: We add this statement in “Ethical statement”.

Reviewer 3 Report

General comments: Overall, very well written manuscript regarding a timely, interesting topic.

Specific comments:

Materials/methods: 

-Page 2 last paragraph discusses left brachial artery access and placement of a 4 french cobra catheter.  Was a microcatheter used to embolize the GDA and/or RGA as discussed?  What happened after this embolization.  Was a pump inserted surgically? If so, a brief summary of that technique should be added.  Was the lipiodol injected via this pump?  

-Was the HAIC regimen used (page 3, first paragraph) from another study and should be cited or was this a new regimen created by the authors?

-What was the imaging protocol for this study?  What modality(s) was used? What was the imaging interval?

Results: 

-Either I have not been provided a figure/table demonstrating radiologic response to treatment or one has not been created.  This is necessary information for the results section and should be included.

-Figure 2 Page 6. Images for 2D and 2E appear to have been inverted.

Discussion:

Nice discussion and literature review.

Page 12, 1st full paragraph discusses "MVI." Please define this acronym.

Author Response

Thank you for your valuable comments.

Followings are our replies:

General comments: Overall, very well written manuscript regarding a timely, interesting topic.

Specific comments:

Materials/methods: 

  1. Page 2 last paragraph discusses left brachial artery access and placement of a 4 french cobra catheter.  Was a microcatheter used to embolize the GDA and/or RGA as discussed?  What happened after this embolization.  Was a pump inserted surgically? If so, a brief summary of that technique should be added.  Was the lipiodol injected via this pump?  
    Reply: In our technique, we placed a temporary 4F angio-catheter in the common or proper hepatic artery for chemoinfusion without implantation of a permanent reservoir system. Usually 4F catheter is used to embolize the GDA. A microcatheter is used in some patients who revealed prominent RGA. The lipiodol is slowing injected manually – added in the subtitle of New Hepatic Arterial Infusion Chemotherapy (HAIC) .
  2. Was the HAIC regimen used (page 3, first paragraph) from another study and should be cited or was this a new regimen created by the authors?
    Reply: This new HAIC regimen was created by ourselves since 2002.
  3. What was the imaging protocol for this study?  What modality(s) was used? What was the imaging interval?

Results: Patients were followed up by “Three-phase computed tomography (CT) or dynamic magnetic resonance imaging (MRI) of the liver after every 2 courses of treatment”-stated in the 3rd paragraph of the subtitle “New Hepatic Arterial Infusion Chemotherapy (HAIC).

,

  1. Either I have not been provided a figure/table demonstrating radiologic response to treatment or one has not been created. This is necessary information for the results section and should be included.
    Reply: We add a table to demonstrate radiologic response as per your suggestion.

-Figure 2 Page 6. Images for 2D and 2E appear to have been inverted.
Reply: The order of Figure 2 seemed not to be inverted. We delete Fig. 2A as per the other reviewer’s suggestion.

Discussion:

  1. Nice discussion and literature review.

Page 12, 1st full paragraph discusses "MVI." Please define this acronym.

    Reply: We add the acronym of MVI “macrovascular invasion” in “Discussion”.

Round 2

Reviewer 3 Report

General comments:

The authors have done a good job of addressing the prior questions/concerns. Thank you.

Specific comments:

What does "DCR" denote in Table 2?  A figure legend with abbreviations to accompany table 2 may be helpful.

Page 13 Final sentence. Please change typographic error of "rasdiologic" to "radiologic"

Page 19  Final paragraph--recommend rewording to "Yttrium 90 radioembolization particles , especially glass microspheres, have a small size (30 um) allowing for deep infiltration.." "The absence of overt arterial embolic effect is important for patients with portal venous occlusion..."